# Prevalence and Risk Factors for Vitamin D Deficiency in Children and Adolescents in the Kingdom of Bahrain

**DOI:** 10.3390/nu15030494

**Published:** 2023-01-18

**Authors:** Buthaina Yusuf Al-Ajlan, Afnan Freije, Sabika Allehdan, Simone Perna

**Affiliations:** 1Nutrition Department in Public Health, Ministry of Health, Sanabis P.O. Box 12, Bahrain; 2Department of Biology, College of Science, UOB Sakhir Campus, University of Bahrain, Sakhir P.O. Box 32038, Bahrain

**Keywords:** vitamin D deficiency, children, adolescent, prevalence Bahrain, risk factors

## Abstract

Background: Vitamin D deficiency has reached pandemic levels in the Middle East and North Africa (MENA) region, even though sunshine is abundant all year round for the cutaneous synthesis of vitamin D through the skin. This study aimed to determine the prevalence of vitamin D deficiency and risk factors associated with serum 25-hydroxy vitamin D (25(OH)D) in children and adolescents aged from 10 to 19 years, as well as the possible associations of vitamin D with calcium, magnesium and phosphate levels. Methods: A multi-center, cross-sectional study was conducted between May and August 2019 at the Ministry of Health in the Kingdom of Bahrain. A total of 383 boys and girls were selected from five health centers from five different regions in the Kingdom of Bahrain. Information about sex, age, education level, weight, height, degree of sunlight exposure, and physical activity levels was recorded. A blood sample was taken from each participant to test serum levels of 25(OH)D, calcium, magnesium and phosphate. Results: The results revealed that 92.1% of the participants were deficient in vitamin D. A significantly higher percentage of boys (96.2%) were vitamin D deficient (<20 ng/mL) than girls (88.3%) (*p* value = 0.004). Vitamin D deficiency were more prevalent among overweight (96.8%) and obese (96.2%) participants than normal body weight and wasted participants (*p* value < 0.001). Being male, overweight, or obese was significantly positively associated with a risk of vitamin D deficiency. Vitamin D deficiency was significantly associated with low serum levels of magnesium. No significant associations were detected between vitamin D deficiency and calcium and phosphate serum levels. However, vitamin D deficiency was significantly associated with low serum level of magnesium (*p* value = 0.017). Conclusions: Our study revealed that vitamin D deficiency was more prevalent among overweight and obese adolescents and mostly boys rather than girls. Magnesium and phosphate were lower in adolescents and children with lower serum 25(OH)D, showing a clear association between these biomarkers and the 25(OH)D.

## 1. Introduction

Hypovitaminosis D is now considered an epidemic, affecting more than a billion people worldwide [1]. This is quite an alarming statistic considering the role of vitamin D in the body and its effect on many of the important physiological processes and its involvement in many neurological and skeletal and non-skeletal diseases [2]. A recent study conducted in Bahrain among healthy children aged 1 month to 16 years reported a high prevalence of vitamin D deficiency (78.3% (<51 nmol/L), whereas 15.1% of them had vitamin D insufficiency (51–74 nmol/L), and only 6.6% had sufficient vitamin D (≥75 nmol/L) [3]. It is well established that the major source of vitamin D is cutaneous synthesis upon the skin’s exposure to natural sunlight, the 7-dehydrocholesterol in the skin absorbs ultraviolet B (UVB) radiation and is transformed into pre-vitamin D3 which then configures into vitamin D3 [4].

Several cutaneous factors affect the synthesis of vitamin D such as skin pigmentation [5], exposure timing and duration [6,7], percentage of area of skin exposed to the sun [7] and sun-protective agents [8]. Moreover, serum vitamin D levels are linked to other factors such as age, gender, ethnicity, seasonal variations and conservative clothing [9,10]. Studies also found that physically active persons who engaged in physical activities on a weekly basis had higher circulating levels of serum vitamin D [11,12]. Vitamin D plays an important role in intestinal absorption of calcium and phosphorus [13]. There are numerous vitamin D-dependent calcium transport proteins found in the intestines, and if vitamin D levels are not adequate, then calcium absorption falls to 10–15% of the dietary intake, in comparison to 30–40% at optimal vitamin D levels. If vitamin D levels are insufficient, then proper mineralization of the bones is compromised [13]. In addition, magnesium increases the bioactivity of vitamin D as it is a cofactor for the vitamin D-binding protein. Thus, the low levels of magnesium results in low serum levels of vitamin D [14]. In the Middle East, many studies have been carried out to estimate the prevalence of vitamin D deficiency and its risk factors among children and adolescents [15,16,17] and only one study conducted among Bahraini children and adolescents [3]. Therefore, vitamin D status in children and adolescents should be excessively investigated due to the impact of vitamin D on their growth and development as well as prevention of other short- and long-term negative clinical consequences associated with vitamin D deficiency such as rickets, hypertension, cardiovascular diseases, chronic musculoskeletal pain and various cancers [18,19]. This study aimed to explore the prevalence and risk factors of vitamin D deficiency in Bahraini children and adolescents aged between 10 and 19 years, as well as the possible associations between vitamin D deficiency and some mineral levels such as calcium, magnesium and phosphate.

## 2. Materials and Methods

### 2.1. Study Participants and Setting

This study was conducted between May and August 2019 at the Ministry of Health premises in the Kingdom of Bahrain. The inclusion criteria of the participants consisted of Bahraini children and adolescents, either boys or girls, whose ages ranged from 10 to 19 years. Participants were excluded if they were known to use any medications that affected bone metabolism, such as any cancer-fighting medications such as chemotherapy or radiation, and other medications such as cortisone.

The selection of health centers was done using a multistage sampling technique. Five health regions were included as five strata including all the health centers pertaining to each health region. One health center was selected from each stratum at random, and participants were enrolled conveniently in this study.

Ethical approval was received from the Ethics Committee of the Ministry of Health, Manama, Kingdom of Bahrain. A signed consent form was obtained from participants’ parents before enrollment in this study. 

### 2.2. Sample Size

The sample size of 389 participants was calculated using the following formula: N = z^2^ (p (1 − p))/d^2^

where N = desired sample size, z = standard normal deviate (set at 1.96, 95% confidence level), D = degree of accuracy desired (0.05) and p = expected prevalence of vitamin D deficiency among adolescents attending health centers (50%) [20].

### 2.3. The Study Questionnaire

The first part of the study was based on a questionnaire taken from the study of Al-Othman et al. [15], which was carried out in the Kingdom of Saudi Arabia. The questionnaire focused on the leading risk factors of vitamin D deficiency and consisted of two sections. The Section 1 focused on the participants’ demographic characteristics which included gender, age and education level. Weight and height were measured by the physician (investigator), alongside the volunteer nutritionists in the health centers. BMI was calculated by dividing weight in kilograms by the square of height in meters. BMI Z-scores were calculated based on age and gender using the World Health Organization growth reference tables [21].

The Section 2 consisted of multiple-choice questions regarding frequency and duration of sun exposure. The responses were then divided into 3 categories: no exposure (<10 min/day), daily exposure (20–30) min/day and weekly exposure (>210 min/week). According to the questionnaire, the daily exposure category means that participants were exposed to UVB light every day, for at least 20 to 30 min. However, participants who were in the weekly category did not get daily sun exposure, and the total amount of minutes under the sun could be divided over 2 to 3 days. 

### 2.4. Physical Activity

Physical activity (PA) was assessed by the interviewer with a face-to-face questionnaire. This questionnaire asked the participants to recall their physical activities (frequency and type of activities performed along with duration—number of minutes per week) [15]. The participants were classified into three categories based on their level of physical activity: inactive, moderately active and active [22]. 

### 2.5. Blood Sample Collection

A 5 mL blood sample was collected from each participant. The blood samples were used to measure serum 25(OH)D, calcium, magnesium and phosphate levels. The chemiluminescence immunoassay analyzer (CLIA) (Abbott Architect i1000sr, Wiesbaden, Germany) was used to quantify levels of 25(OH)D. The coefficient variations for serum 25(OH)D were <13% for values ranged between 8 and 19 ng/mL and <5% for values ranged between 20 and 126 ng/mL. The serum 25OHD assay had specificity for 25(OH) D that was about 91%. Calcium, magnesium and phosphate were measured using ion exchange chromatography.

### 2.6. Statistical Analysis

All statistical analyses were performed using Statistical Package for Social Science (SPSS) version 23.0 (SPSS, Inc., Chicago, IL, USA). Frequencies and percentages were computed for the categorical variable. The chi-square test was used to test the association between categorical variables. Binary logistic regression was used to calculate odd ratios (OR) and their corresponding 95% confidence intervals (CIs) of association of serum (25(OH)D) level with gender, BMI, calcium, magnesium and phosphate on vitamin D deficiency. *p*-value less than 0.05 was considered statistically significant. 

## 3. Results

Participants’ characteristics are summarized in Table 1. A total of 383 participants were enrolled in this study with 4.1% of the original sample size (399) dropped out. The ages of the participants ranged between 10 and 19 years, with a mean of 13.5 years. Many of them were intermediate school students (41.4%), while 25.9% were primary school students, 28.5% were secondary school students and 4.2% were university students. Half of the participants (54.5%) were obese, 25.1% of them were normal body weight, 16.5% were overweight, 2.9% were wasted and only 1% of them were severely wasted. A very large percentage of the participants had vitamin D deficiency at a rate of 92.1%, while 5.8% had insufficient levels of vitamin D, and only 2.1% had sufficient levels. The majority of participants (72.3%) had normal calcium levels, 25.3% had low calcium levels and 2.3% had levels that were too high. About more than half of the participants (58.2%) had low magnesium levels, 39.7% had normal levels and 2.1% had levels that were too high. Most of the participants (74.4%) had normal levels of phosphate, 23.5% had low levels and 2.1% had high levels (Table 1).

Table 2 shows the percentage of participants with regard to exposure to sunlight to the sun and physical activity levels. Regarding participants’ exposure to the sun, 40.7% were exposed to the sun on a daily basis, while 29% had no exposure, and 30.3% had weekly exposure. More than half of the participants (59.8%) were exposed to sunlight around noontime. The second highest percentage of participants (35.4%) were exposed right before sunset, and 4.8% were exposed at sunrise. Only 5.2% were exposed to the sun with their face, and 39.9% were exposed with their face, hands and feet, while the majority (55%) reported that only their face and hands were exposed to sunlight. About 58% of the participants’ clothing covered their entire body, except for their face and hands, while 33.3% of them had clothing that covered their whole body except for their face, hands and feet. The percentages of those who had their whole body covered including the face and whole body except the face were 4.4% and 4.1% of participants, respectively. Furthermore, 86% of the participants did not use any sunscreen, with the remainder of participants (14%) using sunscreen. More than half (65.5%) of the participants stated that they maintained physical activity, while 37.9% of that number were moderately active, and 27.9% were very active (Table 2).

Table 3 presents participant’s characteristics based on serum 25(OH)D levels. A significantly greater percentage of boys (96.2%) were vitamin D deficient than girls (88.3%) (*p* = 0.004). A significantly greater proportion of obese (96.2%) and overweight (96.8%) had low vitamin D compared to normal body weight (82.3%) and wasted (80.0%) participants (*p* < 0.001). It was shown that 95.5% of participants with low magnesium levels were vitamin D deficient, compared to 88.2% and 75% of participants with normal and high levels, respectively (*p* = 0.007). The majority of participants with low (96.7%), normal (91.5%) and high (95.6%) levels of phosphate had vitamin D deficiency (*p* = 0.002).

Table 4 presents the ORs and their 95% CIs of gender, BMI, calcium, magnesium and phosphate on vitamin D deficiency. There was a significant association between gender and vitamin D deficiency. Boys are 3.52 times more likely to have vitamin D deficiency than girls with 95% CI for OR of (1.37, 9.07, *p* = 0.009). In addition, BMI had a significant association with vitamin D deficiency. Overweight participants were 8.56 times more likely to have vitamin D deficiency than normal participants, while the obese participants were 6.29 times more likely to have vitamin D deficiency than normal participants, with 95% CI for OR of (1.54, 47.72, *p* = 0.014) and (2.43, 16.30, *p* <0.001), respectively. Moreover, a low serum level of magnesium was associated positively with vitamin D deficiency. Participants with low magnesium were 2.91-fold more likely to be deficient in vitamin D as compared with participants with normal levels of magnesium, with 95% CI for OR of (1.21,7.01, *p* = 0.017). On the other hand, there were no significant associations between vitamin D deficiency and calcium and phosphate level.

## 4. Discussion

Vitamin D deficiency has become a global concern in children [23]; vitamin D synthesis from the skin upon exposure to sunlight should be possible between 300 to 364 days of the year [24]. This contradiction, especially evident in the Gulf countries, has attracted a lot of attention among researchers who are trying to find out the possible reasons for not having sufficient amounts of serum 25(OH) D levels [24].

In this study, the majority of participants (91.2%) were deficient in vitamin D. Similar findings were reported by Sherief and colleagues [17], who found 94.8% of Egyptian adolescents had vitamin D deficiency. A significant level of vitamin D deficiency was found among boys (96.2%), while 88.3% of girls were deficient. The result of the present study is inconsistent with the study performed in Bahrain by Isa et al. (2020), in which vitamin D deficiency was more prevalent in girls than boys but with no statistical significance [3].

Vitamin D is synthesized in the skin via the reaction of 7-dehydrocholesterol with UVB sunlight exposure, which is considered the main source of vitamin D for the majority of the population, rather than dietary sources. According to the results of this study, more than half of the participants (59.8%) were exposed to sunlight around noon. This is within the general period for the peak of vitamin D synthesis upon UVB exposure [25].

Moreover, a significant association was found between hypovitaminosis D and higher BMI. Obese and overweight participants had 6.29- and 8.56-fold greater risk of vitamin D deficiency, respectively. This finding is consistent with a Spanish study that found BMI was inversely correlated with serum vitamin D levels [26]. This is attributed to a higher fat level in obese and overweight individuals, which in turn reserves the fat-soluble vitamin in the adipose tissue, leaving only small amounts of serum vitamin D in one’s blood circulation [27]. Obesity also changes vitamin D metabolism, for example, the production of the hydroxylation enzymes, which are responsible for activating the inert form of vitamin D, is impaired in obese individuals [25,26,27]. In addition, vitamin D may have a role in the regulation, differentiation and metabolism of adipose tissue [27]. Whether high BMI leads to hypovitaminosis D, or vice versa, is a question that researchers have not found an answer to yet [28].

Lifestyle factors such as dietary intake, physical activity and exposure to sunlight are not considered the only factors responsible for vitamin D levels and status in individuals. Evidence has linked vitamin D status with genetic factors, and there are several candidate genes such as cytochrome P450, family 2, R (CYP2R1) gene; the group-specific component (GC) gene; and the 7-dehydrocholesterol reductase/NAD synthetase 1 (DHCR7/NADSYN1) gene that may contribute to the serum level variability of vitamin D among individuals [29,30]. There are around six single nucleotide polymorphisms (SNPs) situated for vitamin D pathways [31]. Some of these SNPs are involved in vitamin D transport and hydroxylation (from inert vitamin D to the active form), which sequentially increases the risk of vitamin D deficiency in individuals [32]. People from the Gulf countries may have some of these SNPs. For example, one study reported on two Saudi adolescent siblings who were diagnosed with short stature and rickets because they had mutations in their CYP2R1 gene [33].

Vitamin D is well known for its role in calcium homeostasis, bone health and other physiological functions in the human body. So, calcium serum levels correlate with vitamin D levels [13]. The calcium serum levels are much lower for individuals with vitamin D deficiencies. Vitamin D increases calcium absorption in the intestines and the transference of calcium from the bones [34]. The absorption of calcium decreases in the intestines when vitamin D levels are low, and then the amount of ionized calcium available for uptake also declines. The parathyroid glands realize this change and increase parathyroid hormone (PTH) secretion, which keeps levels of calcium serum maintained [13,35].

The prevalence of low serum 25(OH)D levels was high in participants with low (96.7%) or high (95.6%) phosphate levels in this study. This is a major concern as when PTH is high, it also causes the loss of phosphorus into the urine which causes an insufficient amount of calcium-phosphate product, which is needed for the collagen matrix of the bone and its mineralization [35]. This study showed a significant positive association between vitamin D deficiency and low serum magnesium. Likewise, Kelishadi et al. found that serum magnesium associated significantly with 25(OH)D levels [36]. The linear regression analysis revealed that as the Mg level increases by one unit, the 25(OH) D level increases by 0.276 [36]. This relationship may contribute to the central role of magnesium in the activation of vitamin D, which helps regulate mineral homeostasis and parathyroid hormone secretion, which influence bone mineralization [37]. Magnesium is a cofactor for the vitamin D-binding protein and enzymes that activate vitamin D in the liver and kidneys [38]. A high consumption of magnesium was found to decrease the risk of vitamin D deficiency or insufficiency in the general population, as reported by Ahluwalia et al.

The main limitation of this study is that other factors that affect vitamin D levels were not investigated, for instance, clothing, indoor lifestyle, geographical location, skin pigmentation, dietary intake of vitamin D, seasonal variation and environmental pollution.

In spite of this limitation, the main strength of this study was that the prevalence of vitamin D deficiency was determined using a sample of children and adolescents from five health regions in Bahrain, which means the results may be representative of the whole population of Bahrain. Moreover, the study provides further evidence to public health policy makers to propose interventions to improve vitamin D status in Bahrain, particularly for children and adolescents.

## 5. Conclusions

Our study revealed that vitamin D deficiency is a common health problem among Bahrain children and adolescents aged between 10 and 19 years, affecting boys more than girls.

In particular, vitamin D deficiency was more prevalent among overweight and obese adolescents. Magnesium and phosphate were lower in adolescents and children with lower vitamin D, showing a clear association between these biomarkers and vitamin D.

Although our results provide some evidence that obesity is associated with vitamin D deficiency, the effects of some environmental factors on the UBV levels in Bahrain should also be considered. It might be appropriate to consider enhancing the oral intake of vitamin D for practical and cultural reasons.

## Figures and Tables

**Table 1 nutrients-15-00494-t001:** Participant characteristics (*n* = 383).

		(Mean ± SD)
Age		13.5 ± 2.6
		*n* (%)
Gender	Boy	186 (48.6)
	Girl	197 (51.4)
Educational level	Primary	99 (25.9)
	Intermediate	158 (41.4)
	Secondary	109 (28.5)
	University	16 (4.2)
BMI	Normal	96 (25.1)
	Overweight	63 (16.5)
	Obese	208 (54.5)
	Wasted	11 (2.9)
	Severe Wasted	4 (1)
Vitamin D	Deficient	352 (92.1)
	Insufficient	22 (5.8)
	Sufficient	8 (2.1)
Calcium	Normal	277 (72.3)
	Low	97 (25.3)
	High	9 (2.3)
Magnesium	Normal	152 (39.7)
	Low	223 (58.2)
	High	8 (2.1)
Phosphate	Normal	285 (74.4)
	Low	90 (23.5)
	High	8 (2.1)

*n* = total number of participants. Vitamin D categories: deficient < 20 ng/mL, insufficient = 20–29 ng/mL, sufficient ≥ 30 ng/mL. Calcium categories: Low < 2.10–2.55 < High. Magnesium categories: Low < 0.74–1 < High. Phosphate categories: Low < 0.95–1.65 < High. BMI was calculated according to the WHO BMI-for-age boys’, and BMI-for-age girls’ growth charts [16].

**Table 2 nutrients-15-00494-t002:** Exposure to the sun and physical activity (*n* = 383).

		*n* (%)
To what extent do you get exposed to the sun?	No exposure	111 (29)
	Daily	156 (40.7)
	Weekly	116 (30.3)
Which time of the day do you get exposed to the sun?	Sun rise	13 (4.8)
	Noon	162 (59.8)
	Before sunset	96 (35.4)
Which body parts mostly get exposed to the sun?	Face	14 (5.2)
	Face and hands	149 (55)
	Face, hands, and feet	108 (39.9)
To what extent do you cover your body while getting exposed to the sun?	All body except face and hands	157 (58.1)
	All body except face	11 (4.1)
	All body except face, hands, and feet	90 (33.3)
	All body including face	12 (4.4)
Do you use sun protection creams?	Yes	38 (14)
	No	233 (86)
Do you maintain physical Activity?	Yes	251 (65.5)
	No	132 (34.5)
Activity level	Inactive	131 (34.2)
	Moderate	145 (37.9)
	Active	107 (27.9)

*n* = total number of participants. No exposure: no minutes under the sun. Daily exposure: every day, 20–30 min under the sun. Weekly exposure: a total of >210 min under the sun in total, not daily. Inactive: no movement. Moderately active: 60 min of physical activity daily, <20% high intensity. Vigorously active: 60 min of physical activity daily, >20% high intensity.

**Table 3 nutrients-15-00494-t003:** Vitamin D in relation to participant characteristics.

		Vitamin D		Chi-Squared *p*-Value
		Not Deficient	Deficient	
		*n* (%)	*n* (%)	
Age	≤10	3 (4.4)	65 (95.6)	0.306
	11–15	17 (7.6)	206 (92.4)	
	16–19	10 (11)	81 (89)	
Gender	Boy	7 (3.8)	178 (96.2)	0.004
	Girl	23 (11.7)	174 (88.3)	
Educational level	Primary	7 (7.1)	92 (92.9)	0.495
	Intermediate	11 (7)	146 (93)	
	Secondary	11 (10.1)	98 (89.9)	
	University	0 (0)	16 (100)	
BMI	Normal	17 (17.7)	79 (82.3)	<0.001
	Overweight	2 (3.2)	60 (96.8)	
	Obese	8 (3.8)	200 (96.2)	
	Wasted	3 (20)	12 (80)	
Calcium	Normal	23 (8.3)	253 (91.7)	0.635
	Low	7 (7.2)	90 (92.8)	
	High	0 (0)	9 (100)	
Magnesium	Normal	18 (11.8)	134 (88.2)	0.007
	Low	10 (4.5)	212 (95.5)	
	High	2 (25)	6 (75)	
Phosphate	Normal	24 (8.5)	260 (91.5)	0.002
	Low	3 (3.3)	87 (96.7)	
	High	3 (4.4)	65 (95.6)	

*n* = Number of participants. Vitamin D deficiency < 20 ng/mL. Vitamin D insufficiency = 20–29 ng/mL. Vitamin D sufficiency = 30–100 ng/mL.

**Table 4 nutrients-15-00494-t004:** Binary logistic regression of gender, BMI, calcium, magnesium and phosphate for R vitamin D deficiency.

	*p*-Value	OR	95% CI for OR	
			Lower	Upper
Gender				
Girls	Reference			
Boy	0.009	3.52	1.37	9.07
BMI				
Normal	Reference			
Overweight	0.014	8.56	1.54	7.72
Obese	0.000	6.29	2.43	16.30
Wasted	0.485	0.59	0.13	2.62
Calcium				
Normal	Reference			
Low	0.305	0.59	0.21	1.63
High	-------	-----	-----	-----
Magnesium				
Normal	Reference			
Low	0.017	2.91	1.21	7.01
High	0.154	0.23	0.03	1.74
Phosphate				
Normal	Reference			
Low	0.244	2.30	0.57	9.36
High	0.163	0.27	0.04	1.69

OR: odd ratio. 95% CI: 95% confidence interval.

## Data Availability

Not applicable.

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
