# Peer review of "Prevalence and Risk Factors for Vitamin D Deficiency in Children and Adolescents in the Kingdom of Bahrain"

_nutrients, 2023, doi:10.3390/nu15030494_

Round 1

Reviewer 1 Report (Previous Reviewer 2)

I am attaching my edits here. The study needs an extensive grammatical/typographical review before it is fit to publish.

Thanks.  

Author Response

Dear Editor:

We would like to thank you and the reviewers for taking the time to carefully review our manuscript and we sincerely appreciate their helpful suggestions to improve the quality of our manuscript. Following the reviewers’ concerns and comments, we made all needed modifications to the initial version of our manuscript which we marked by red color.

Please find our responses to their suggestions. Please note that we made the responses to the reviewers in blue color as well as we mentioned the line number and page number.

With many thanks.

The authors

Reviewer 1

The authors did not measure vitamin D in serum - - so replace “vitamin D serum levels “ by serum 25OHD....

Response: It has been changed. Line 13, 33, 34, 122, 123.

The abstract does not mention the threshold for vitamin D deficiency – so the 92% prevalence is meaningless without that information ! please change the wording--- I recommend adding also the mean or median concentration –  this referee found out that deficiency is  less than 200 ng/ml but this was mentioned only in the legend of a table.

Response:  We are totally agree with your comment, but unfortunately data about serum 25 hydroxy vitamin D, calcium, magnesium and phosphate were categorical not continuous, we received data about these parameters from medical lab as  categories with reference ranges that we mentioned in legend of tables. We mentioned the cut point  of vitamin D deficiency as (< 20 ng/ml) in abstract.  Page 1 Line 23.

This referee is not aware of Magnesium being “cofactor for vitamin D binding protein” – please explain or correct.

Response: It has been added this information. Page 9 Line 265-266.

 Is the lab responsible for the 25OHD using QC samples to evaluate possible bias for accuracy – this should be mentioned including the % bias

Response: Yes, it was.  The serum 25OHD assay had the coefficient variation (CV)  < 13% for value ranged between 8 and 19 ng/ml and < 5% between 20 and 126 ng/ml. The serum 25OHD assay had specificity for 25 OH Vitamin D that about 91%.  Page 3 Line 124-127.

 Bahrein is a country with a large number of (temporary) immigrants with different lifestyle – are only Bahrein nationals included?

Response: It had mentioned in the inclusion criteria that enrolledc participants were Bahraini children and adolescents either boys or girls whose ages ranged from 10 to 19 years. Page 2 Line 76-77.

 Table 4 would be much more informative if a column with absolute values of 25OHD would be  included

Response:  We are totally agree with your comment, but unfortunately data about serum 25 hydroxy vitamin were categorical non numerical.

 The information about Mg concentrations are far to superficial to evaluate its relevance

Response: Data about Magnesium were categorical non numerical.

The relevance of the data would be greatly enhanced by adding data on either PTH or    Alk Phase or both – a bone endpoint would even be b

Response:  The data about PTH and Alk Phase are not available as well as we didn't have enough fund to measure the PTH and Alk Phase.

Reviewer 2 Report (New Reviewer)

Comments

1.      The authors did not measure vitamin D in serum - - so replace “vitamin D serum levels “ by serum 25OHD....

2.      The abstract does not mention the threshold for vitamin D deficiency – so the 92% prevalence is meaningless without that information ! please change the wording--- I recommend adding also the mean or median concentration –  this referee found out that deficiency is  less than 200 ng/ml but this was mentioned only in the legend of a table

3.      This referee is not aware of Magnesium being “cofactor for vitamin D binding protein” – please explain or correct

4.      Is the lab responsible for the 25OHD using QC samples to evaluate possible bias for accuracy – this should be mentioned including the % bias

5.      Bahrein is a country with a large number of (temporary) immigrants with different lifestyle – are only Bahrein nationals included?

6.      Table 4 would be much more informative if a column with absolute values of 25OHD would be  included

7.      The information about Mg concentrations are far to superficial to evaluate its relevance

The relevance of the data would be greatly enhanced by adding data on either PTH or    Alk Phase or both – a bone endpoint would even be b

Author Response

Reviewer 2

Line 17 was selected => were selected

Response: It has been corrected.

Line19 sun => sunlight

Response: It has been corrected.

Line 25 Vitamin D significantly associated with low serum levels of Magnesium. Vitamin D deficiency was significantly associated with low serum levels of Magnesium.

Response: It has been corrected.

Line 21 - 27 A significantly higher percentage of boys (96.2%) were vitamin D deficient than girls (88.3%) (P value = 0.004). Vitamin D deficiency were more prevalent among overweigh (96.8%) and obese (96.2%) participants than normal body weight and underweight participants (P value < 0.001). Can you please combine the statements above with the one below to give the abstract a better focus please? Also, there were significant positive associations between vitamin D serum levels and gender (P value < 0.05) and Body mass index (BMI) (P value < 0.001).

Response: The required change has been done.

Line 38 Hypovitaminosis D is now considering as an epidemic affected more than a billion people worldwide change to Hypovitaminosis D is now considered an epidemic, affecting more than a billion people worldwide

Response: the statement has been changed.

Line 42 from

Response: It has been deleted.

Line 44 insufficient => insufficiency

Response: It has been corrected.

Line 45 established that the major

Response: It has been corrected.

 Line 49 Several cutaneous factors that affect the

Response: It has been corrected.

Line 53 physical active person => physically active persons

Response: It has been corrected.

Line 63 [15- 18]and => [15–18] and

Response: It has been corrected.

 Line 64 On => one

Response: It has been corrected.

 Line 70 adolescents who aged 77 who ages => whose ages

Response: It has been corrected.

Line 86 …constant? Please check

Response: It has been corrected.

Line137 10 and 19 years of age, with

Response: It has been corrected.

Line141 …thinness? Please check

Response: It has been changed to wasted.

Line155 156 exposure to the sun => exposure to sunlight

Response: It has been corrected.

Line 157 158 160 sun => sunlight

Response: It has been corrected.

Line 163,164 …of them clothing covered their whole body => …of them had clothing that covered

Response: It has been corrected.

Line 167 (14%) used => (14%) who used

Response: It has been corrected.

Line 201 Table 4, please check font type used, maintain consistency.

Response: It has been reformatted. 

Line 211 994% ….note Sherief reports 94.8% 213 boy participants (96.2%) => boys participants (96.2%)

Response: It has been corrected.

Line 217, 218 Incomplete sentence

Response: It has been corrected.

Line 224 with Spanish => with a Spanish

Response: It has been corrected.

Line 226 …reserved => …reserves

Response: It has been corrected.

Line 228 …for example, production => …for example, the production

Response: It has been corrected.

Line 233 answer, yet => answer yet

Response: It has been corrected.

Line 244 … hemostasis => …haemostasis

Response: It has been corrected.

Line 245 …body, So, calcium => …body. So, calcium

Response: It has been corrected.

Line 254 This is major concern => This is a major concern

Response: It has been corrected.

Line 261 contributed to central => contributed to the central

Response: It has been corrected.

Line 263 Magnesium is cofactor => Magnesium is a cofactor

 Response: It has been corrected.

Line 264, 265 A high consumption of magnesium decreased the risks of vitamin D deficiency or insufficiency in the general population was reported by Ahluwalia et al. [39]. A high consumption of magnesium was found to decrease the risk of vitamin D deficiency or insufficiency in the general population as reported by Ahluwalia et al. [39].

 Response: It has been done.

Line 268, 269 Please rephrase this statement. If you refer to the instrument you used in the data collection that it is not reliable and not valid, what is the point of using it then? Alternatively please delete this sentence.

 Response: It has been done.

Line 280 …deficiency is common => ..deficiency is a common

Response: It has been corrected.

Line 281,282 …between 10 and 19 years. Boys were more affected than girls. …between 10 and 19 years, affecting boys more than girls.

Response: It has been corrected.

Line 283 Overweigh => overweight

 Response: It has been done.

Line284 Adolescents and mostly boys rather than girls.

Response: They have been deleted.

Line 287, 288 Although our results provided some evidence that the obesity and serum levels of magnesium associated vitamin D deficiency. The effects of some environmental factors on the UBV levels in Bahrain should also be considered.

Although our results provide some evidence that obesity is associated with vitamin D deficiency, the effects of some environmental factors on the UBV levels in Bahrain should also be considered

Response: It has been done.  

This manuscript is a resubmission of an earlier submission. The following is a list of the peer review reports and author responses from that submission.

Round 1

Reviewer 1 Report

In this study, the authors investigated the prevalence and risk factors associated with vitamin D deficiency in children and adolescents in Bahrain. This is a descriptive epidemiological study and cross-sectional study. Lack of novelty of the research question, limited contribution to the advancement of science, and deficits in the robustness of methods used include the drawbacks of this study.  I wish to highlight the following concerns that should be addressed before this article can be considered for further steps:

1.     I recommend this manuscript be reviewed by an individual with professional proficiency in English to correct errors in grammar, punctuation, word choice, and sentence construction to ensure that the document reads as through written by a Native English speaker.

2.    Several typographical errors throughout the manuscript (for example, missing parentheses, content-based errors, and extra periods, among others) should be corrected.

3.    The manuscript should be revised in accordance with the Strengthening the Reporting of Observational Studies in Epidemiology (STROBE) statement.

4.    (Line 64): This is a cross-sectional study; therefore, authors should avoid words such as “effects” that indicate causal relationships.

5.    (Line 78): Specific p (expected prevalence of vitamin D deficiency) and calculated sample size should be added to the manuscript.

6.    (Line 92): It is necessary to clarify the rationale for use of the World Health Organization body mass index (BMI) for the age growth reference table to calculate the BMI. It is important to state whether the authors calculated the Z score.

7.    (Line 101): The authors need to provide details regarding evaluation of the PA.

8.    (Line 111): It is necessary to clarify the expression "with a correlation coefficient of 0.92."

9.    (Line 113): Details regarding logistic regression analysis should be added to the Methods section.

10. (Figure 1): In my opinion, data will be easier to understand if heatmaps are replaced with simple concrete numbers.

11. It is necessary to describe the limitations of this study in accordance with STROBE guidelines.

Author Response

Dear Editor:

We would like to thank you and the reviewers for taking the time to carefully review our manuscript and we sincerely appreciate their helpful suggestions to improve the quality of our manuscript. Following the reviewers’ concerns and comments, we made all requried modifications to the initial version of our manuscript which we marked by red color.

Please find our responses to their suggestions. Please note that we made the responses to the reviewers in red color as well as we mentioned the line number and page number.

With many thanks.

The authors

UTHOR´S RESPONSES TO REVIEWERS´ COMMENTS

Dear Editor:

We would like to  thank   you and   the reviewers   for  taking the   time  to  carefully  review  our

manuscript and we sincerely appreciate their helpful suggestions to improve the quality of our

manuscript. Following the reviewers’ concerns and comments, we made some modifications to

the initial version of our manuscript which we marked by track changes. We also rewrite some

paragraphs in order to reduce similarity.  

Please find our responses to their suggestions. Please note that we made the responses to the

reviewers in blue color as well as we mentioned the line number.

With many thanks.

The auth

Reviewer 1

In this study, the authors investigated the prevalence and risk factors associated with vitamin D deficiency in children and adolescents in Bahrain. This is a descriptive epidemiological study and cross-sectional study. Lack of novelty of the research question, limited contribution to the advancement of science, and deficits in the robustness of methods used include the drawbacks of this study.  I wish to highlight the following concerns that should be addressed before this article can be considered for further steps:

  1. I recommend this manuscript be reviewed by an individual with professional proficiency in English to correct errors in grammar, punctuation, word choice, and sentence construction to ensure that the document reads as through written by a Native English speaker.

Response: It has been reviewed by expert in English language

  1. Several typographical errors throughout the manuscript (for example, missing parentheses, content-based errors, and extra periods, among others) should be corrected.

Response: All typographical errors have been corrected.

  1. The manuscript should be revised in accordance with the Strengthening the Reporting of Observational Studies in Epidemiology (STROBE) statement.

 Response: It has been revised based on STROBE.

  1. (Line 64): This is a cross-sectional study; therefore, authors should avoid words such as “effects” that indicate causal relationships.

         Response : It has been corrected. Lines 73-77 page 3

  1. (Line 78): Specific p (expected prevalence of vitamin D deficiency) and calculated sample size should be added to the manuscript.

          Response : It has been mentioned. Lines 98-102 page 4

  1. (Line 92): It is necessary to clarify the rationale for use of the World Health Organization body mass index (BMI) for the age growth reference table to calculate the BMI. It is important to state whether the authors calculated the Z score.

Response: It has been reported. Lines 109-112, page 4

  1. (Line 101): The authors need to provide details regarding evaluation of the PA.

Response: It has been clarified and mentioned. “Physical activity was assessed by face-to-face interview questionnaire. This questionnaire asked the participants to recall there (frequency and type of activities performed along with duration – number of minutes-per week). The subjects were divided into three groups based on their level physical activity, subjects were divided into three groups: inactive, moderately active and active [15]”. Page: 4; Line 125-130.

  1. (Line 111): It is necessary to clarify the expression "with a correlation coefficient of 0.92."

Response: It has been removed; the correlation coefficient is not related to the mentioned equipment.  Page: 4, Line 134.

  1. (Line 113): Details regarding logistic regression analysis should be added to the Methods section.

Response: It has been added, Page: 5, Line 142-145.

  1. (Figure 1): In my opinion, data will be easier to understand if heatmaps are replaced with simple concrete numbers.

Response: Vitamin D was correlated only with vitamin D and correlation scale was represented in right side of figure 1. Page: 9

  1. It is necessary to describe the limitations of this study in accordance with STROBE guidelines.

Response: It has been mentioned as well as strength of our study.  Page: 11, Line:  308-320.

Author Response

Reviewer 2

  1. Lines 34, 37, 42–45, 50, 63, 208, 210, 217, 220, 238, 246, 263, 272, 297, 307, 317, 327, 331 Please check punctuations. , in place of . or . before or after [ref].

Response: They have been modified.

  1. Line 2 47 Content 25(OH)D. First mention please define

Response: It has been done.

  1. Line 79 Content Change “patients” to “participants”

Response: It has been done.

  1. Line 85 Content “study of” to “study by”

Response: It has been done.

  1. Line 88 Typo “consist” – “consisted”

Response: It has been done.

  1. Line 18 Content: Is this qualitative variable or quantitative variables

Response: They are qualitative variables

  1. Lin 108 Exactly 40.7% of the participants were exposed to the sun on a daily basis, while 29% had no exposure, and 30.3% had weekly exposure. Use of “Exactly” in this sentence is not appropriate. Would you rephrase to a sentence like below: Regarding participant’s exposure to the sun, 40.7% were exposed to the sun on a daily basis, while 29% had no exposure, and 30.3% had weekly exposure.

Response: It has been corrected.

  1. Line 160 Content Table 2 For the variables: Do you use sun protection creams? And Do you maintain physical Activity? Present the “Yes” row only, delete the row for “No” it is redundant.

Response: It has been corrected.

  1. Line 192 Grammar BMI has significant effect To BMI was found to have a significant effect.

Response: It has been corrected.

  1. Line193 Typo “are” to “were” please revise the whole section.

Response: It has been corrected.

Line 200 Typo The variables on the left most column seems to have shifted up by one slot. Please check.

Response: It has been modified.

  1. Line 251 Typo took part of physical activity To took part in physical activity.

Response: It has been corrected.

  1. Line 289 Typo “intestinal” to “intestines”

Response: It has been corrected.

  1. Line 331 Typo “his” to “This”

Response: It has been corrected.

Round 2

Reviewer 1 Report

Thank you for your detailed responses to each of my comments.

However, considering the major issue, the novelty of this research question, and its contribution to the advancement of science, it still does not fully meet the criteria for an original research article in this journal.